# Estimates of the incidence, prevalence, and factors associated with common sexually transmitted infections among Lebanese women

Hiam Chemaitelly[1,2], Ramzi R. Finan[3], Eddie Racoubian[4], Gulzhanat Aimagambetova[5], Wassim Y. Almawi[6,7] *

1 Infectious Disease Epidemiology Group, Weill Cornell Medicine-Qatar, Cornell University, Doha, Qatar, 2 Department of Population Health Sciences, Weill Cornell Medicine, Cornell University, Ithaca, New York, United States of America, 3 Department of Obstetrics and Gynecology, Hôtel Dieu de France, CHU Université St. Joseph, Beirut, Lebanon, 4 St. March Medical and Diagnostic Center, Beirut, Lebanon, 5 Department of Surgery, School of Medicine, Nazarbayev University, Astana, Kazakhstan, 6 Department of Biological Sciences, Brock University, St. Catharines, Ontario, Canada, 7 Faculty of Sciences, El Manar University, Tunis, Tunisia

* wassim.almawi@outlook.com

**Data Availability Statement:** All relevant data are within the paper and its Supporting Information files.

## Abstract

### Background

We analyzed the prevalence of active infection with common curable sexually transmitted infections (STIs) including *N. gonorrhea*, *C. trachomatis*, *T. vaginalis*, and *T. pallidum*, as well as active infection with HPV, herpes simplex virus types I (HSV-1) and II (HSV-2), *M. hominis*, *M. genitalium*, *C. albicans*, and Ureaplasma in 351 Lebanese women.

### Methods

A cross-sectional study, involving 351 sexually active women, 40 years or younger, who were recruited from outpatient Obstetrics and Gynecology clinic attendees between September 2016 and November 2017.

### Results

The prevalence of active infection was low at 0.3% for *N. gonorrhea*, 0.6% for HSV-2, 2.8% for *C. trachomatis*, and 2.9% for any curable STIs. Prevalence of active HPV infection was high assessed at 15.7% for high-risk and 12.2% for low-risk genotypes. Furthermore, the prevalence was 2.0% for *M. genitalium*, 6.8% for ureaplasma, 13.7% for *Candida albicans*, and 20.5% for *M. hominis*. No active infections with *T. vaginalis*, *T. pallidum*, or HSV-1 were observed. Significant age differences were noted in the prevalence of high-risk and low-risk HPV genotypes, but no such differences were noted in the prevalence of other infections. No appreciable variations were identified in the prevalence of key STIs based on smoking, marital status, or the number of sexual partners.

**Funding:** The author(s) received no specific funding for this work.

**Competing interests:** The authors have declared that no competing interests exist.

## Conclusions

The study documented active infection with substantial prevalence for multiple STIs among women attending outpatient gynecology and obstetrics clinics in Lebanon. These findings underscore the importance of strengthening STI surveillance, linkage to care, and prevention interventions in reducing STI incidence among women.

## Introduction

Sexually transmitted infections (STIs) remain a global pervasive public health challenge, with women across wide age groups, and socioeconomic and geographical backgrounds, bearing a significant portion of this burden [1, 2]. STIs encompass an array of bacterial, viral, and parasitic infections that are transmitted through sexual contact, and range from the common chlamydia and gonorrhea to the more serious and likely life-threatening infections such as HIV/AIDS [2, 3]. Several factors are associated with the persistence of STIs, which include changes in sexual behavior, lack of sufficient awareness, evolving pathogens, inadequate access to healthcare, and stigma [4, 5]. STIs differ in their scope, characteristics, prognosis, and consequences, which can vary from localized inflammatory disorder to cancer, and even death [4, 6], thus necessitating comprehensive STI investigation and testing in women at a young age, and periodically thereafter.

Infection with STIs can result in a range of associated complications if left untreated [7, 8]. These include cervical cancer, infertility, chronic pain, and heightened neonatal morbidity and mortality [2, 9]. Accurate and timely detection of active STIs is crucial for the affected individual and public health management [9, 10]. The advent of DNA/RNA-based PCR has allowed for highly sensitive and specific detection of STI pathogens in different sample types, including blood, urine, and vaginal discharges [9, 11]. Despite the availability of ultra-sensitive diagnostic tests for STI causative agents, the prevalence of STIs remains a public health concern, both in developed, developing, and under-developed countries [2, 6].

Lebanon is situated in the East Mediterranean, and its present-day population comprise a mix of Christian (Catholic/Maronite, Greek Orthodox, Copts) and Moslem (Sunni, Shiite, Druze) communities [12, 13]. Several socio-cultural factors including limited sex education, gender inequality, social norms and beliefs, and high-risk sexual behaviors, along with healthcare issues and challenges, comprising limited access to and cost of healthcare, deteriorating healthcare system, and lack of comprehensive STI services contribute to the increase in the STI prevalence [14]. This is compounded by the recent refugee crisis [15, 16], emerging STI strains, and limited data collection and surveillance, thus highlighting the need for a comprehensive approach aimed at providing access to affordable healthcare services and strengthening STI surveillance [14, 16].

This study investigated the prevalence of active infection with *N. gonorrhea*, *C. trachomatis*, *T. vaginalis*, *T. pallidum*, Human papillomavirus (HPV), herpes simplex virus types I (HSV-1) and II (HSV-2), *M. hominis*, *M. genitalium*, *C. albicans*, and ureaplasma in 351 Lebanese women attending outpatient obstetrics and gynecology (OB/GYN) clinics, of whom 291 were 40 years or younger. The study further explored the variations in prevalence by age, smoking status, marital status, and number of sexual partners.

## Subjects and methods

### Study participants

The study participants comprised 351 women who were attending the outpatient OB/GYN clinics at Hôtel-Dieu de France Hospital and St. Marc Medical Center in the Greater Beirut

area, between September 1, 2016, and November 30, 2017. All women presented to the clinic for routine gynecological check-up and were asymptomatic, and none reported pregnant at the time she was invited to participate in the study. A written informed consent was obtained from consenting participants.

## Data collection

Patients' socio-demographic and risk behavior data were retrieved through a review of medical charts, and a unified questionnaire that details the demographic characteristics, sexual behaviors, history of STIs, and other relevant factors. The confidentiality of the participants was strictly maintained throughout the study, which included procedures for data storage handling and access, and anonymization.

## Specimens' collection

Endocervical specimens and first-catch urine specimens were then collected and processed within 4 hours of collection. Swabs were collected by a gynecologist using speculum-assisted Spatula of Ayre after removing the mucus and were placed in a balanced saline solution. Total genomic DNA was extracted by the mini-spin column method (Qiagen, Hilden, Germany). DNA from urine specimens was extracted by the Roche Amplicor STD specimen preparation kit (Roche Diagnostics, Mannheim, Germany), according to the manufacturer's specifications.

## HPV DNA amplification

HPV testing on endocervical DNA samples was done by nested PCR using MY09 and MY11 external primers and GP05+/GP06+ internal primers, as shown elsewhere [7]. HPV genotypes were detected by HPV Quant-21® (DNA Technology, Moscow, Russian Federation), and were grouped into low-risk (LR) (HPV6, 11, 42, 43, 44, and 70), and high-risk (HR) (HPV16, 18, 26, 31, 33, 35, 39, 45, 51, 52, 53, 56, 58, 59, 66, 68, 73, and 82). Detection of other STIs was also done using a specific DNA Technology Detection kit (DNA Technology).

## Statistical analysis

Participant characteristics were summarized using both frequency distributions and measures of central tendency. In cases where specific variables had missing values (constituting $\leq 2.2\%$ of the dataset), these values were imputed using the median of observed outcomes for individuals with complete data. Descriptive statistics were performed on all testing samples, and the prevalence of active infections, along with the corresponding 95% confidence intervals (CI), were estimated. Stratified analyses were conducted to investigate the distribution of active infections across various factors, including age, smoking status, marital status, and number of sexual partnerships. Chi-square tests were performed to assess the association of each covariate with active infection. A p-value below 0.05 indicated a strong association with active infection. Due to the limited number of PCR-positive samples for *N. gonorrhea*, *C. trachomatis*, *T. vaginalis*, and *T. pallidum* (syphilis), these infections were combined to define the presence of any curable STI. To gain deeper insights into the determinants of active infection with key STIs, exploratory univariate and multivariate regression analyses were performed. However, these analyses had insufficient statistical power to detect statistical significance. All statistical analyses were conducted using Stata/SE version 17.0 (Stata Corporation, College Station, TX, USA).

### Ethical statement

This study was performed in line with the principles of the Declaration of Helsinki. Approval was granted by the Research and Ethics Committee of St. Marc Medical Center (Number, SMMC-2019-0047), granted on October 17, 2019. This study was reported according to the Strengthening the Reporting of Observational Studies in Epidemiology (STROBE) guidelines (S1 Table).

## Results

### Study population

The study included 351 women who underwent testing for an active STI infection. Table 1 shows the characteristics of the study participants. The median age was 33.0 years, with an interquartile range (IQR) of 29.0–37.0 years. Approximately 70% were married, and the rest were single, divorced, or separated. A third of women identified as smokers. About two-thirds reported up to one sexual partner, whereas 15.9% had two sexual partners, and 25.4% had three or more sexual partners.

**Table 1. Characteristics of study participants.**

| Characteristics | Categories | N (%) |
|---|---|---|
| Total sample size | | 351 (100.0) |
| Age (years) | Median (IQR) [1] | 33.0 (29.0–37.0) |
| Age—years | 20–29 years | 100 (28.5) |
| 20–29 years | 30–39 years [2] | 191 (54.4) |
| 30–39 years* | 40–49 years | 49 (14.0) |
| 40–49 years | 50+ years | 11 (3.1) |
| Smoking | | 110 (31.2) |
| Marital | Married | 240 (68.4) |
| | Single/Divorced/Separated [3] | 111 (31.6) |
| Sexually active | | 334 (95.2) |
| Numbers of partners | 0–1 partner [4] | 206 (58.7) |
| | 2 partners | 56 (15.9) |
| | 3+ partners | 89 (25.4) |
| HSV-1 | | 0 (0.0) |
| HSV-2 | | 2 (0.6) |
| *Neisseria gonorrhoeae* | | 1 (0.3) |
| *Chlamydia trachomatis* | | 10 (2.8) |
| *Trichomonas vaginalis* | | 0 (0.0) |
| *Treponema pallidum* | | 0 (0.0) |
| *Mycoplasma hominis* | | 72 (20.5) |
| *Mycoplasma genitalium* | | 7 (2.0) |
| HPV-High risk | | 55 (15.7) |
| HPV-Low risk | | 43 (12.2) |
| *Candida albicans* | | 48 (13.7) |
| Ureaplasma | | 24 (6.8) |

**IQR**, interquartile range.

1. 8 observations were imputed at the median age of 33 years.

2. Includes 6 missing observations.

3. Includes 7 missing observations.

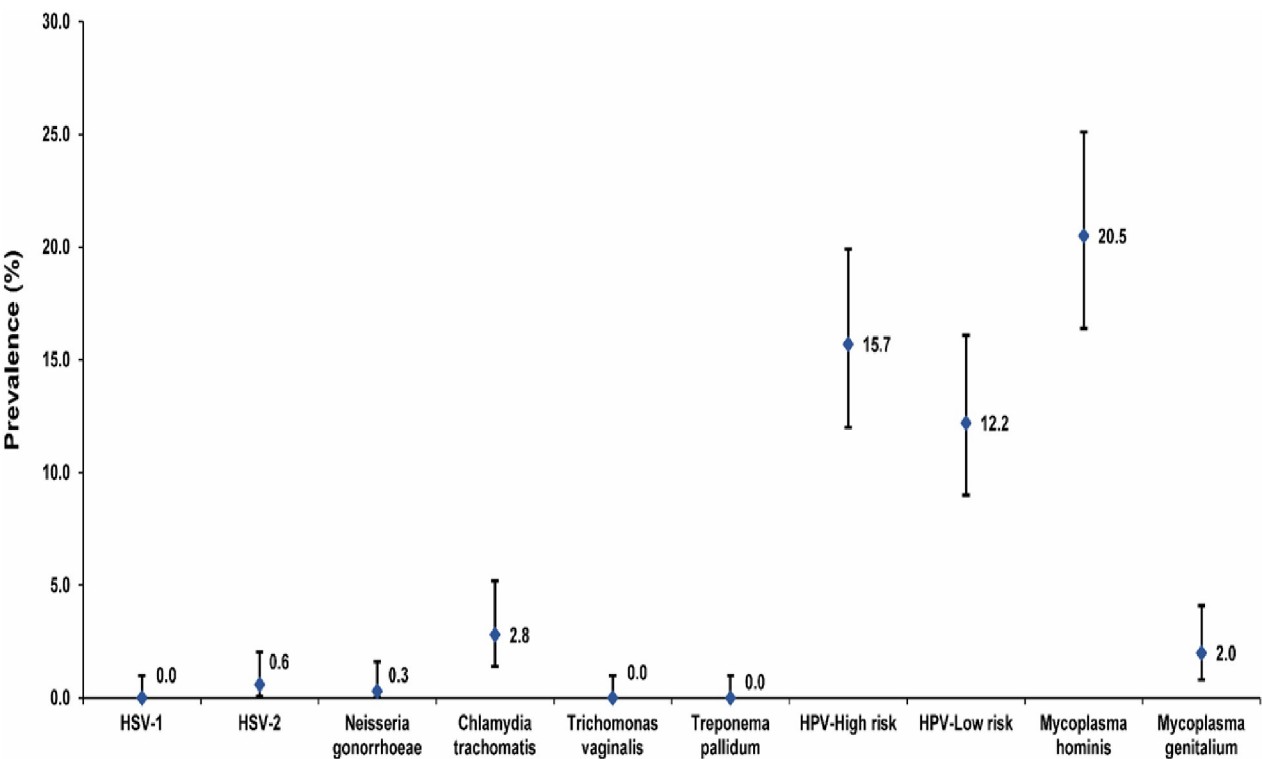

**Fig 1. Prevalence of active infection with sexually transmitted infections among women gynecology clinic attendees in Beirut, Lebanon.**

### Active infection prevalence

The prevalence of active infections among study participants (as identified by PCR) was nil (95% CI: 0.0–1.0%) for HSV-1 and low at 0.6% (95% CI: 0.1–2.0%) for HSV-2, indicating minimal instances of active viral shedding (Fig 1). While the prevalence for any curable STI was 2.9% (95% CI: 1.4–5.2%), the prevalence for curable bacterial STIs was 0.3% (95% CI: 0.0–1.6%) for *N. gonorrhea*, 2.8% (95% CI: 1.4–5.2%) for *C. trachomatis*, and nil (95% CI: 0.0–1.0%) for each of *T. vaginalis* and *T. pallidum*. Active HPV infection was common with a prevalence of 15.7% (95% CI: 12.0–19.9%) for HR genotypes and 12.2% (95% CI: 9.0–16.1%) for LR genotypes. Prevalence was 2.0% (95% CI: 0.8–4.1%) for *M. genitalium* and 20.5% (95% CI: 16.4–25.1%) for *M. hominis*, 6. Compared to the rates of 8% (95% CI: 4.4–10.0%) seen for Ureaplasma and 13.7% (95% CI: 10.3–17.7%) for *Candida albicans* (Fig 1).

### Patterns of active infection

Fig 2 illustrates the distribution of active infection prevalence by age, smoking status, marital status, and number of sexual partners, and detailed statistical associations are provided in S2–S4 Tables. There were negligible variations in prevalence across age groups for *M. genitalium* and for any curable STI, with prevalence ranging from 1.0–2.6% (p-value: 0.632) and 2.0–3.3% (p-value: 0.831), respectively (Fig 2A). A similar pattern was observed for *M. hominis* in the face of the higher prevalence rates, which ranged from 16.7–22.0% (p = 0.665). Meanwhile, significant age differences were noted in the prevalence of HR HPV, which was lowest in women aged 20–29 years (7.0%) and highest in women aged 30–39 years (20.9%) (p = 0.007). Prevalence of LR HPV also varied in women aged 20–29 years (7.0%) compared to 30–39-year-old women (15.2%) (p = 0.128).

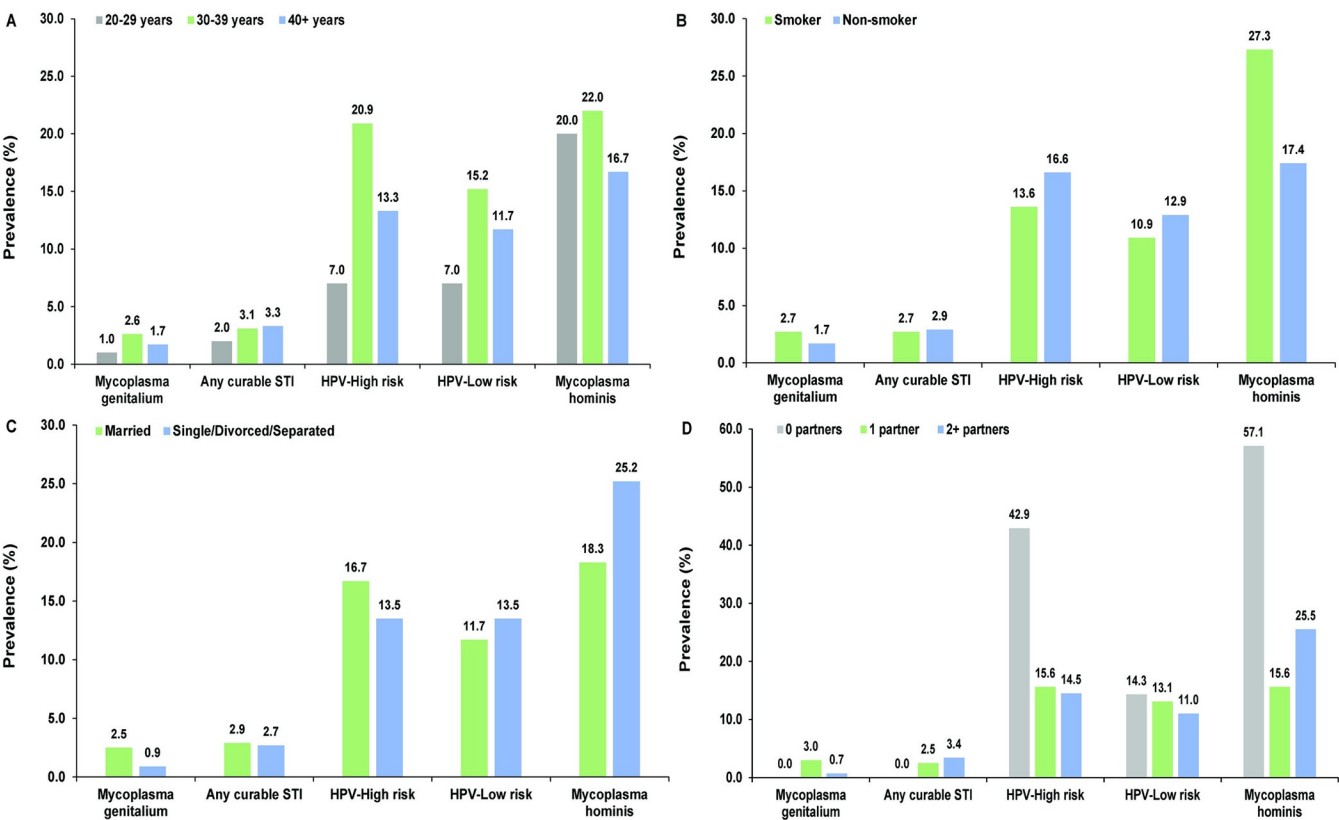

Any curable STI was defined as an active infection with any of *Neisseria gonorrhoeae*, *Chlamydia trachomatis*, *Trichomonas vaginalis*, and T*reponema pallidum*.

**Fig 2.** Prevalence of active infection with sexually transmitted infections A) by age group, B) smoking status, C) marital status, and D) number of sexual partners among women gynecology clinic attendees in Beirut, Lebanon. Any curable STI was defined as an active infection with any of *Neisseria gonorrhoeae*, *Chlamydia trachomatis*, *Trichomonas vaginalis*, and *Treponema pallidum*.

No appreciable differences were observed in the prevalence of key STIs by smoking status (Fig 2B), except for *M. hominis* where the prevalence was 27.3% among smokers compared to 17.4% among non-smokers (p-value: 0.034). Negligible differences in prevalence were also noted by marital status across all STIs (Fig 2C), despite the prevalence of *M. hominis* being much higher in single, divorced, and separated women (25.2%) compared to married women (18.3%). Minimal variation in the prevalence of key STIs was also found for women reporting none, one, and two or more sexual partners (Fig 2D). The prevalence among women reporting 0 partners reflects the underreporting of partnerships for social desirability. These results confirm the findings of our main analysis of substantial STI prevalence that goes undetected among women regardless of the reported number of partners. Of note, the prevalence of *M. hominis* was substantially higher in the latter (25.5%) compared to the former group (17.0%; p-value: 0.051).

## Discussion

Our assessment of active STI infections in a group of women attending OB/GYN clinics revealed prevalence levels for common STIs that align with global estimates. These prevalence levels are however higher than expected, given the generally conservative sexual norms in the Middle East and North Africa (MENA) region. Specifically, the prevalence of *N. gonorrhoeae* and *C. trachomatis* were 0.3% and 2.8%, respectively, within the CIs of the global prevalence

for these infections estimated by the WHO at 0.8% and 3.2%, respectively [17]. The prevalence of HR (15.7%) and LR (12.2%) HPV genotypes were also considerable, comparable to levels observed in Asia, but lower than levels observed in developed countries such as Europe and North America where routine screening for HPV is implemented [18].

The higher-than-expected prevalence identified among women may not necessarily imply increased engagement in sexual risk behaviors [14, 16]. Instead, these findings underscore the lack of sexual health services and the limited capacity for STI prevention and treatment in Lebanon, and the broader MENA region [15, 19]. Socio-cultural sensitivities, constrained resources, and competing national health priorities have contributed to STIs being neglected in research and surveillance, as well as on national public health agendas [19, 20]. Consequently, infections often go undetected, persisting for prolonged periods, and heightening the potential for infection circulation within the population [16, 19]. This situation significantly elevates the risk of future health complications among women [19, 21].

The minimal variations in active infection levels by smoking status, marital status, and the number of lifetime sexual partners affirm the notion that insufficient access to STI services is the primary factor contributing to the higher-than-expected prevalence levels [14]. In particular, the prevalence among women reporting 0 partners reflects the underreporting of number of sexual partnerships for social considerations (stigma and shame) and taboos surrounding sexuality. These results strongly suggest substantially higher STI prevalence that goes undetected among women regardless of the self-reported number of partners. Our findings showed variability in HR and LR HPV across age groups, with the most elevated prevalence found among women aged 30–39 years [22]. This pattern could signify an increased likelihood of virus exposure over time. However, it may also be influenced by a survival bias, where women in older age groups, having a greater chance of progressing to cervical cancer, are less likely to have survived the complications associated with HPV. These findings are consistent with existing literature indicating the highest prevalence of HPV in this age group, as well as the increased incidence of cervical cancer among older women [23, 24].

There is a pressing need to integrate HPV screening into routine assessments for women of reproductive age in Lebanon and other Middle East and North Africa region (MENA) countries or, at the very least, to enhance awareness about the significance of HPV screening among women [25], akin to the campaigns established for breast cancer awareness [14, 25, 26]. Several cultural, economic, and access to healthcare factors contribute to the rising, but not fully documented, prevalence of STIs [6, 19, 21]. These include cultural and religious norms that lead to reluctance to seek testing and treatment for STIs, and conservative attitudes in those communities, which limit the ability of women to negotiate safer sex practices [20, 25, 30]. Socioeconomic factors, coupled with the cost of and limited access to healthcare services, and an emerging influx of refugees have widened the disparities in access to healthcare and efficient STI services [15, 16, 19, 20].

Co-infections of HPV-infected women with other STIs, namely Chlamydia, Mycobacteria, and HIV, were linked to HPV persistence, and increased cervical neoplasia risk [4, 27]. Insofar as they were reported for different ethnic groups and pathophysiological conditions [3, 28], this suggested a geographical pattern of co-infection [10, 29], likely linked with socioeconomic and education status and inadequate access to healthcare facilities [5, 7, 30]. The majority of women with co-infections were positive for HPV (8.6%) as part of double infection with mostly *M. hominis*, which is attributed to the high prevalence of these two STIs globally. This was in line with findings on Nigerian women, in which persistent *M. hominis* infection was linked with HPV positivity, without establishing a clear cause-and-effect link [31]. Other coinfections included mycoplasma with chlamydia and candida, which were not as pronounced as

HPV coinfections, in agreement with a recent German study [30]. However, co-infection prevalence was independently associated with smoking and having multiple sexual partners [4, 7].

## Strengths and limitations

This study has limitations. The term "co-infection" was based on the detection of nucleic acid material without indication of the actual disease status. Women were recruited from outpatient clinics in a hospital located in the capital city, and therefore our findings may not be generalizable to the broader female population in the country [32]. The number of identified positive cases was relatively small hindering the conduct of meaningful regression analyses. The limited sample size restricted analysis of complete stratification based on the number of sexual partners. Despite these shortcomings, this study provided evidence of the current prevalence of a broad spectrum of infections among women in the general population, shedding light on a neglected disease burden.

## Conclusions

In conclusion, our study revealed the higher-than-expected prevalence of STIs among women attending OB/GYN clinics in the MENA region, emphasizing the significance of reinforcing STI surveillance to develop a better understanding of the burden of these infections nationwide and monitor infection trends over time [17]. Our findings also underscore the imperative for targeted prevention interventions, including HPV immunization [26], enhanced access to sexual health services, integration of STIs into routine screening protocols, and improved linkage to care [14, 25]. Implementing these strategies is essential to reducing STI incidence and mitigating their adverse impact on women's health, social well-being, and economic welfare.

## Supporting information

**S1 Checklist. Human participants research checklist.**
(DOCX)

**S1 Data.**
(XLSX)

**S1 Table. STROBE guidelines.**
(DOCX)

**S2 Table. Associations with active infection with any of *Neisseria gonorrhoeae*, *Chlamydia trachomatis*, *Trichomonas vaginalis*, and *Treponema pallidum*.**
(DOCX)

**S3 Table. Associations with active infection with human papillomavirus (HPV).**
(DOCX)

**S4 Table. Associations with active infection with *Mycoplasma hominis* and *Mycoplasma genitalium*.**
(DOCX)

## Acknowledgments

The authors are grateful for the technical assistance of the St. Marc Medical Center Molecular Diagnostics Unit.

## Author Contributions

**Conceptualization:** Ramzi R. Finan, Wassim Y. Almawi.

**Data curation:** Ramzi R. Finan, Eddie Racoubian.

**Formal analysis:** Hiam Chemaitelly, Wassim Y. Almawi.

**Investigation:** Eddie Racoubian.

**Methodology:** Hiam Chemaitelly, Eddie Racoubian, Gulzhanat Aimagambetova.

**Resources:** Eddie Racoubian.

**Supervision:** Wassim Y. Almawi.

**Validation:** Hiam Chemaitelly, Ramzi R. Finan, Gulzhanat Aimagambetova.

**Writing – original draft:** Hiam Chemaitelly, Gulzhanat Aimagambetova.

**Writing – review & editing:** Wassim Y. Almawi.

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
