## [Decision Letter · Decision Letter 0]

26 Feb 2024

PONE-D-24-03237Estimates of the incidence, prevalence, and factors associated with common sexually transmitted infections among Lebanese womenPLOS ONE

Dear Dr. Almawi,

Thank you for submitting your manuscript to PLOS ONE. After careful consideration, we feel that it has merit but does not fully meet PLOS ONE’s publication criteria as it currently stands. Therefore, we invite you to submit a revised version of the manuscript that addresses the points raised during the review process.

We look forward to receiving your revised manuscript.

Kind regards,

Angelica Espinosa Miranda, M.D., Ph.D.

Academic Editor

PLOS ONE

Journal Requirements:

3. In the online submission form, you indicated that [The datasets generated and/or analyzed during the current study are available on reasonable request from the principal investigator (WY Almawi) upon request.]. 

Reviewers' comments:

Reviewer's Responses to Questions

**Comments to the Author**

1. Is the manuscript technically sound, and do the data support the conclusions?

Reviewer #1: Yes

Reviewer #2: Partly

2. Has the statistical analysis been performed appropriately and rigorously? 

Reviewer #1: Yes

Reviewer #2: Yes

3. Have the authors made all data underlying the findings in their manuscript fully available?

Reviewer #1: Yes

Reviewer #2: Yes

4. Is the manuscript presented in an intelligible fashion and written in standard English?

Reviewer #1: Yes

Reviewer #2: Yes

5. Review Comments to the Author

Reviewer #1: Congratulations to the authors for their insightful exploration of a critical yet underexplored subject – the prevalence of STIs among Lebanese women. This research brings much-needed attention to a crucial public health issue in a region where such studies are limited, providing valuable insights that can inform targeted interventions and policies.

When it comes to specific suggestions for improvement, in the introduction, mentioning the specific situation in Lebanon would be highly relevant, as the article focuses on STI prevalence among Lebanese women. The author could address the socio-cultural factors, healthcare infrastructure, and challenges in Lebanon that contribute to the observed STI prevalence. This would provide a more comprehensive and localized perspective on the issue.

Moving to the "Subject and Methods" section, consider dividing "Study Subjects" into two distinct subtopics: "Study Participants" and "Data Collection." This structural adjustment will enhance the organization of information, providing readers with a clearer delineation of the study's participant details and the data collection methodology.

To enhance the Results section, begin with a brief summary outlining the key findings, highlighting the overall prevalence of STIs and any noteworthy trends. Additionally, organize the presentation of prevalence rates in a clearer structure by grouping related rates together, possibly using subsections or a table. This adjustment will improve readability and the logical flow of information for the reader.

In the Discussion section, deepen the interpretation of prevalence rates by exploring cultural, economic, and healthcare access factors specific to Lebanon. While the brief mention of higher-than-expected prevalence rates is noted, a more detailed comparative analysis with global estimates and other regions would be interesting. Provide a more detailed comparison of your findings with other relevant studies, especially those conducted in the Middle East and North Africa region. Strengthen this section by emphasizing specific interventions and strategies for improvement. Regarding geographical patterns, although mentioned, further exploration of their implications and recommendations for addressing healthcare disparities would enhance this section.

On page 8, when mentioning "HR and LR HPV," ensure that abbreviations and acronyms are defined upon first use for clarity.

Your study holds significant promise, and these suggested improvements only aim to enhance its impact. Your efforts in addressing these aspects will undoubtedly contribute to the advancement of knowledge in this important field. Keep up the great work!

Reviewer #2: I consider the paper relevant but need some explanation about the topics below.

The article does not clarify whether the tested women attended the gynecology outpatient clinic with complaints (symptoms or clinical signs) or just for routine gynecological check-ups (preventive). There is also no confirmation whether among the examined women there pregnant women and any reference to the gestational period of these ones were. This fact could impact the discovery of STIs by the research.

I suggest to the authors that they separate the group of women without sexual partners from those with one partner and those with two or more sexual partners. Otherwise, it is necessary to detail in the materials and methods because the separation was not made. I also understand it is necessary to establish unequivocally the period reflected by the number of sexual partners, as only using the expression " lifetime sexual partners " in the discussion is not sufficient.

I am curious about the information regarding whether there is a statistically significant difference between the median age of the participants and a positive result for Neisseria gonorrhoeae, Chlamydia trachomatis, Trichomonas vaginalis, and Treponema pallidum.

6. PLOS authors have the option to publish the peer review history of their article (what does this mean?). If published, this will include your full peer review and any attached files.

Reviewer #1: No

Reviewer #2: **Yes: **Edilbert Pellegrini Nahn Junior

---

## [Author Response · Author response to Decision Letter 0]

7 Mar 2024

Reviewer #1: 

Reviewer: This research brings much-needed attention to a crucial public health issue in a region where such studies are limited, providing valuable insights that can inform targeted interventions and policies.

Authors: We believe so, thank you for your comment.

Reviewer: …. in the introduction, mentioning the specific situation in Lebanon would be highly relevant ….. The author could address the socio-cultural factors, healthcare infrastructure, and challenges in Lebanon that contribute to the observed STI prevalence. 

Authors: Will do.

Changes: The Introduction was modified as suggested by the Reviewer. Additional supporting references were added.

Reviewer: ….. consider dividing Study Subjects into two distinct subtopics: Study Participants and Data Collection. This structural adjustment will enhance the organization of information, providing readers with a clearer delineation of the study, participant details and the data collection methodology.

Authors: We acknowledge (and appreciate) this suggestion.

Changes: As suggested by the Reviewer, the Study Subjects section was divided into “Study Participants” and “Data Collection”.

Reviewer: To enhance the Results section, begin with a brief summary outlining the key findings, highlighting the overall prevalence of STIs and any noteworthy trends. Additionally, organize the presentation of prevalence rates in a clearer structure by grouping related rates together, possibly using subsections or a table. This adjustment will improve readability and the logical flow of information for the reader.

Authors: Noted.

Changes: The Results section was restructured as suggested by the Reviewer.

Reviewer: In the Discussion section, deepen the interpretation of prevalence rates by exploring cultural, economic, and healthcare access factors specific to Lebanon, …. a more detailed comparative analysis with global estimates and other regions would be interesting …… especially those conducted in the Middle East and North Africa region…… emphasizing specific interventions and strategies for improvement. Regarding geographical patterns, further exploration of their implications and recommendations for addressing healthcare disparities would enhance this section.

Authors: We thank the Reviewer for this suggestion.

Changes: The Discussion section was modified as per the comment of the Reviewer.

Reviewer: On page 8, when mentioning HR and LR HPV; ensure that abbreviations and acronyms are defined upon first use for clarity.

Authors: Noted.

Changes: HR, LR and other abbreviations and acronyms were checked for definition before use.

Reviewer #2: 

Reviewer: The article does not clarify whether the tested women attended the gynecology outpatient clinic with complaints (symptoms or clinical signs) or just for routine gynecological check-ups (preventive). There is also no confirmation whether among the examined women there pregnant women and any reference to the gestational period of these ones were. 

Authors: Participating women reported to the clinic for routine check-up, and none of the participating women was pregnant at the time she entered the study.

Changes: The “Study Subjects” section of the Methods was appropriately modified.

Reviewer: I suggest to the authors that they separate the group of women without sexual partners from those with one partner and those with two or more sexual partners, it is necessary to detail in the materials and methods ….. it is necessary to establish unequivocally the period reflected by the number of sexual partners, as only using expression “lifetime sexual partners” in the discussion is not sufficient.

Authors: We thank the reviewer for the comment. We did conduct a stratified analysis to separate women with higher sexual risk behaviour (defined as ≥2 sexual partners) from those with lower sexual risk behaviour (defined as none or only one sexual partner) for two reasons. The first is the unavoidable underreporting of the actual number of sexual partners by women, a phenomenon documented among unmarried women in less conservative settings such as sub-Saharan Africa (see Omori et al., Sexually Transmitted Infections 2015;91:451-7). The second relates to the limited sample size which hinders stratifying the entire study findings by partnership status.

Changes: The Results section, including modification to Figure 2, along with appropriate changes in the Discussion, were made.

Reviewer: I am curious about the information regarding whether there is a statistically significant difference between the median age of the participants and a positive result for Neisseria gonorrhoeae, Chlamydia trachomatis, Trichomonas vaginalis, and Treponema pallidum.

Authors: There was no statistical difference between the median age of women testing positive for any curable STI, assessed at 34 years and that for women testing negative for any curable STI, assessed at 33 years (p-value 0.663).

Changes: Appropriate changes where made where indicated.

---

## [Editor Report · Decision Letter 1]

13 Mar 2024

Estimates of the incidence, prevalence, and factors associated with common sexually transmitted infections among Lebanese women

PONE-D-24-03237R1

Dear Dr. Almawi,

We’re pleased to inform you that your manuscript has been judged scientifically suitable for publication and will be formally accepted for publication once it meets all outstanding technical requirements.

Kind regards,

Angelica Espinosa Miranda, M.D., Ph.D.

Academic Editor

PLOS ONE

---

## [Editor Report · Acceptance letter]

28 Mar 2024

PONE-D-24-03237R1 

PLOS ONE

Dear Dr. Almawi, 

I'm pleased to inform you that your manuscript has been deemed suitable for publication in PLOS ONE. Congratulations! Your manuscript is now being handed over to our production team.

Kind regards, 

on behalf of

Dr. Angelica Espinosa Miranda 

Academic Editor

PLOS ONE